# RASSF1A Tumour Suppressor: Target the Network for Effective Cancer Therapy

**DOI:** 10.3390/cancers12010229

**Published:** 2020-01-17

**Authors:** Lucía García-Gutiérrez, Stephanie McKenna, Walter Kolch, David Matallanas

**Affiliations:** 1Systems Biology Ireland, University College Dublin, Belfield, Dublin 4, Ireland; lucia.garcia@ucd.ie (L.G.-G.); stephanie.mc-kenna@ucdconnect.ie (S.M.); walter.kolch@ucd.ie (W.K.); 2School of Medicine, University College Dublin, Belfield, Dublin 4, Ireland; 3Conway Institute, University College Dublin, Belfield, Dublin 4, Ireland

**Keywords:** RASSF1A, tumour suppressor, cancer, therapy, apoptosis and Hippo pathway, DNMTP

## Abstract

The RASSF1A tumour suppressor is a scaffold protein that is involved in cell signalling. Increasing evidence shows that this protein sits at the crossroad of a complex signalling network, which includes key regulators of cellular homeostasis, such as Ras, MST2/Hippo, p53, and death receptor pathways. The loss of expression of RASSF1A is one of the most common events in solid tumours and is usually caused by gene silencing through DNA methylation. Thus, re-expression of RASSF1A or therapeutic targeting of effector modules of its complex signalling network, is a promising avenue for treating several tumour types. Here, we review the main modules of the RASSF1A signalling network and the evidence for the effects of network deregulation in different cancer types. In particular, we summarise the epigenetic mechanism that mediates RASSF1A promoter methylation and the Hippo and RAF1 signalling modules. Finally, we discuss different strategies that are described for re-establishing RASSF1A function and how a multitargeting pathway approach selecting druggable nodes in this network could lead to new cancer treatments.

## 1. Introduction

RASSF1A, which is a well described tumour suppressor, is one of the most commonly deregulated genes in human cancers [1]. RASSF1A is a scaffold protein member of the C-terminal RASSF family [2] and is encoded by the *RASSF1* gene [2,3]. This gene expresses eight isoforms due to alternative splicing, and RASSF1A along with RASSF1C are the most abundantly expressed isoforms. RASSF1A is a signalling hub that plays an important role in signal transduction, where it seems to mediate a complex network that includes the ERK, Hippo, apoptotic and p53 subnetworks [3]. RASSF1A is also a downstream effector of KRAS mediating the antiproliferative signal that is triggered by the activation of this proto-oncogene [4]. The RASSF1A signalling network (SN) can mediate the regulation of different biological functions, such as apoptosis, cell cycle arrest, regulation of the microtubule network, migration, and autophagy [3,5,6]. Unsurprisingly, the deregulation of RASSF1A function results in the development of different pathologies, including cancer and cardiovascular diseases [7]. However, we still have a very patchy picture of the RASSF1A SN (particularly its upstream regulators) and how this tumour suppressor inputs different signals to effector pathways. 

The study of RASSF1A at the molecular level and the characterisation of the mechanistic properties of its signalling network are complicated by the very nature of scaffold proteins. These proteins do not have enzymatic functions and they exert their biological effect by bringing their effector proteins together [8,9]. The assembly of these protein complexes increases the local concentration of signalling nodes, which can accelerate biochemical reactions and also compartmentalise signalling into specific protein complexes or subcellular localisations. Importantly, under physiological conditions, scaffold proteins have an optimal concentration of expression, where the stoichiometry of the substrates and the scaffold lead to the complete assembly of scaffolded proteins, thereby optimising the biochemical reaction rates between them (Figure 1) [9,10]. The expression of the scaffold protein below or above this optimal concentration window results in an inhibition or suboptimal regulation of the molecular reactions mediated by the bound effectors [11]. Importantly, changes in expression levels of the scaffold proteins are associated with pathological deregulation of signalling networks in several diseases [12]. In the case of RASSF1A, the most common mode of deregulation is the loss of expression by gene promoter hypermethylation [2]. Additionally, it is possible that, in several tumours, RASSF1A is still expressed, but at suboptimal (due to loss of heterozygosity of the 3p21.3 region [3]) or even supra-optimal concentration. In all of these scenarios, the deregulation of RASSF1A will lead to a defective regulation of its signal transduction network, which seems to be a common event in cancer development [13,14,15,16]. In this review, we give an overview of the current knowledge of the RASSF1A SN with special consideration for how the different subnetworks may contribute to the tumour suppressor function of RASSF1A. Of note, we refer to signalling modules rather than classical effector pathways and to the individual proteins in the network as nodes in order to give a systems level description for the SN. We specifically focus on the different strategies that have been tested in pre-clinical and clinical models to rescue RASSF1A deregulation. Finally, we discuss potential therapeutic strategies that may be developed to target the different modules of the RASSF1A SN.

## 2. Regulation of the RASSF1A Protein

As a scaffold protein, the RASSF1A functions are mediated by protein-protein interaction with its effectors (Figure 1). Changes in RASSF1A protein expression and post-translational modifications (PTM) affect the formation of these protein complexes and regulate the activation of the RASSF1A SN [17]. 

### 2.1. Epigenetic Regulation of RASSF1A Expression

DNA methylation of two CpG islands located at its promoter regulates *RASSF1A* expression (Figure 2A and recently reviewed by Malpeli et al [18]). This methylation is mediated by DNA methyltransferases, such as DNMT1, DNMT3A, and DNMT3B [19,20,21]. While methylation of the *RASSF1A* promoter has been shown in cells without affecting the expression of the gene, hypermethylation results in *RASSF1A* loss of expression [18]. This seems to occur in normal cells and it has been related to physiological processes, such as ageing [22]. Unfortunately, very little is known about the physiological regulation of RASSF1A expression by methylation, though it is likely mediated by cell signalling. However, aberrant hypermethylation of the *RASSF1A* promoter is one of the most common events in cancer and it has been shown to be caused by the deregulation of DMNTs [19,20,21]. Thus, in pathological conditions, such as cancer and viral infection, hypermethylation of the *RASSF1A* promoter results in the deregulation of its complex SN and is thought to be necessary for the development of disease. Interestingly, *RASSF1A* promoter hypermethylation seems to result in the transcription of its splicing isoform RASSF1C, which can bind and regulate several RASSF1A effectors with opposite outcomes [23]. Recent evidence has shed light on the mechanisms that regulate the pathogenic hypermethylation of *RASSF1A*. Kufe’s group has shown that MUC1-C binds to the *RASSF1A* promoter and recruits the co-repressor ZEB1 [24]. The MUC1-C-ZEB1 complex recruits DNMT3B, which methylates the CpG island located at the *RASSF1A* promoter. It is possible that the same mechanism regulates the physiological expression of *RASSF1A*, but further experiments are necessary for determining this. This process has also been mirrored by p53 binding to the *RASSF1A* promoter, which results in DAXX recruitment [25]. DAXX recruits DMNT1 to the *RASSF1A* promoter, leading to hypermethylation of its CpG islands and the loss of expression. As already mentioned, viral infection also induces the loss of expression of *RASSF1A* by promoter hypermethylation. For example, Hepatitis virus (HV) A, B, and C induce methylation in hepatocytes affecting a great number of tumour suppressor genes, including *RASSF1A*, which has shown high association with the development of hepatocellular carcinoma (HCC) [26,27,28]. The mechanisms that regulate HV-dependent DNA methylation are not completely understood, but they are likely to be mediated by inflammatory signals and direct regulation of DNMTs by viral proteins, such as HBx [29]. Additionally, important evidence links *RASSF1A* hypermethylation to human papilloma virus (HPV) infection-dependent cervical carcinoma and it might prevent the activation of p53 antioncogenic signals [30,31,32,33]. Importantly, the silencing of *RASSF1A* by hypermethylation has been shown to occur at the same time as promoter hypermethylation of the *TP73* gene, which is an effector of RASSF1A [34,35]. This concomitant loss of expression of RASSF1A and p73 proteins can be induced by HPV and associated with HCC. Finally, *RASSF1A* expression has also been shown to be regulated by Epstein-Barr Virus (EBV) infection, where it is linked to B-Cell transformation [36]. In this case, the viral protein EBNA3C promotes DNMT expression and activity, which results in *RASSF1A* promoter hypermethylation. EBV-dependent loss of expression of *RASSF1A* might also be related to the development of other cancer types, such as nasopharyngeal carcinoma [37]. 

### 2.2. RASSF1A Regulation by Post-Translational Modifications (PTM)

Several PTMs have been shown to regulate RASSF1A functions. RASSF1A is phosphorylated at different residues [28,38,39], which changes RASSF1A affinity to some of its effectors (Figure 2B). Mechanistically, the best characterised phosphorylation affects the RASSF1A S131, which has been shown to be phosphorylated by ATM in response to DNA damage [39]. This phosphorylation promotes the interaction of RASSF1A with MST2, leading to cell cycle arrest. The observation that a nonsynonymous single nucleotide polymorphism (A133S), which prevents the phosphorylation of the S131, seems to increase resistance to radiotherapy in some patients with soft tissue sarcoma, highlights the importance of this residue [40]. Additionally, the phosphorylation of RASSF1A by GSK3-β on several residues (S175, S178, and S179) was shown to promote its interaction with 14-3-3 proteins preventing RASSF1A interaction with several pro-apoptotic proteins [41]. PKC has also been shown to phosphorylate RASSF1A at S193 and S203 residues preventing RASSF1A activity in the reorganisation of the microtubule network [42]. PKC dependent phosphorylation of RASSF1A has recently been shown in hypoxic conditions, resulting in stabilisation of RASSF1A and its binding to HIF1α, the master regulator of the hypoxic response [43]. Similarly, Aurora A kinase regulates RASSF1A by phosphorylating the residues T202 and S203 in the Ras association domain, which leads to the loss of RASSF1A association with microtubules during mitosis [44,45]. Importantly, RASSF1A phosphorylation does not only regulate RASSF1A function by modulating protein-protein interaction, but they also regulate protein stability. In this regard, the phosphorylation of RASSF1A S203 by Aurora A kinase is needed for RASSF1A to be recognised by the APC ubiquitination complex [46]. As a result, RASSF1A, is degraded in late mitosis, allowing for cell cycle progression. The importance of RASSF1A phosphorylation and the control of its protein levels by ubiquitination was further supported by results from Lim’s group showing that CDK4 also phosphorylates RASSF1A at S203 during G1-S transition. In this case, RASSF1A phosphorylation allows the binding of Skp2 a subunit of the Skp1-Cul1-F-box (SCF) ubiquitin ligase complex targeting RASSF1A for proteasomal degradation, leading to cell cycle progression [38]. It must be noted that RASSF1A ubiquitination might occur independent of phosphorylation, as indicated by the observation that Cullin-4A, which is part of the CUL4A-DDB1-RING complex, scaffolds the interaction of RASSF1A with DDB1, and subsequent ubiquitination by this E2 ligase [47]. Finally, RASSF1A interaction with TGFβ allows for the E3 ubiquitin ligase ITCH to ubiquitinate RASSF1A, which promotes its degradation and the translocation of YAP to the nucleus [48]. In summary, these examples clearly demonstrate the importance of PTMs in the regulation of RASSF1A signalling. As many of the proteins that regulate RASSF1A PTMs are commonly deregulated in tumour cells [49], they may be the key to understanding how the deregulation of the RASSF1A SN leads to cell transformation. 

## 3. RASSF1A Signalling Network

There are several reviews that focus on the current knowledge of the RASSF1A SN [2,3,17,50]. In this section, we want to summarise the main modules of the RASSF1A SN with a focus on the possible treatments for cancers, where this protein is deregulated (summarised in Figure 3).

### 3.1. Extracellular Signals and Upstream Regulators of RASSF1A

Very few extracellular signals have been demonstrated to regulate RASSF1A. Importantly, RASSF1A signalling is regulated by pro-apoptotic insults, which activate both intrinsic and extrinsic apoptotic modules. Thus, RASSF1A is a mediator of the death receptor (DR) family and has been shown to be regulated by the DRs FAS, TNFR, and TRAILR [51,52,53,54]. Upon the activation of these receptors by their ligands TNFα, TRAIL, or FASLG, RASSF1A can interact with some of the proteins of the death-inducing signalling complex (DISC) and mediate the activation of the extrinsic apoptotic module. In particular, RASSF1A has been shown to bind to DAXX, one of the DISC proteins, which results in the regulation of p53 transcriptional activity [55]. The activation of the DRs also recruits MOAP-1 to the DISC mediating the interaction of this protein with RASSF1A [53]. MOAP-1 undergoes a conformational change when in complex with RASSF1A that results in the activation of this pro-apoptotic protein. Activated MOAP-1 has been shown to trigger the extrinsic apoptotic module by regulating the BCL2 proteins [56]. Finally, several groups have shown that the DRs promote the interaction of RASSF1A with MST1/2 Ser/Thr kinases, which are core kinases of the Hippo pathway [51,57,58]. However, it is unknown as to whether this complex requires the participation of DISC proteins [6]. ATM and ATR are other well-known activators of the RASSF1A pro-apoptotic signal [39]. These kinases are activated by the DNA damage response and, therefore, link RASSF1A with the intrinsic apoptotic module. 

RASSF1A signalling is also regulated by mitogenic stimuli, which activate receptor tyrosine kinases (RTKs), such as the epidermal growth factor receptor (EGFR) [59]. These signals control RASSF1A effects on the regulation of cell cycle progression and microtubule organisation [5]. RASSF1A is a negative regulator of these functions and it has been shown to impede cell cycle progression through different mechanisms. Importantly, KRAS seems to be the main activator of RASSF1A upon mitogenic stimulation [4] and RASSF1A has been shown to mediate the pro-apoptotic signals that have been described for the KRAS oncogene. As such, RASSF1A has opposite effects to the best-characterised effectors of KRAS, RAF kinases, and PI3K. In particular, we have shown that the EGFR can promote the interaction of RASSF1A with MST2, but it prevents the pro-apoptotic signal mediated by MST2 [59]. Thus, EGFR regulates the crosstalk of the Hippo and RAF-MEK-ERK modules. Additionally, the EGFR activates the PI3K module, which results in the activation of AKT [60]. This kinase further contributes to the inhibition of MST2 by direct phosphorylation in a RASSF1A dependent fashion [60]. Therefore, mitogenic stimuli prevent RASSF1A pro-apoptotic signals. However, constant activation of RAS results in the activation of the RASSF1A pro-apoptotic signal due to EGFR or KRAS mutation [48,59]. 

Calcium has also been shown to regulate RASSF1A signalling [61]. It was shown that PMCA4b, which is a plasma membrane Ca2+ pump, interacts with RASSF1A. Interestingly, this binding resulted in the inhibition of the EGFR-dependent activation of ERK, further demonstrating the role of RASSF1A in the regulation of KRAS effectors. Unfortunately, this observation from Neysis’ group was not further investigated and very little is known regarding the role of PMCA4 in the RASSF1A SN. Since recent data have shown that RASSF1A plays a key role in heart homeostasis [62] where calcium signalling, which is central in the regulation of myocyte physiology, [63] might prove to be a key regulator of RASSF1A function. 

### 3.2. Downstream RASSF1A Effector Signalling Network

The previous sections already demonstrate that RASSF1A regulates a complex SN, which is still being characterised. In this section, we want to briefly summarise the best studied downstream effector modules. For extended reviews see references [2,3,5,50,64].

#### 3.2.1. Canonical Apoptotic Pathways

As mentioned above, RASSF1A directly activates the extrinsic apoptotic pathway downstream of the DRs through the regulation of MOAP-1 in a RAS dependent fashion [65]. Upon TNFR signalling, RASSF1A is released from 14-3-3 proteins and can recruit MOAP-1, relieving this latter protein from auto-inhibition. Subsequently, the RASSF1A-MOAP-1 complex mediates the activation of BAX by direct interaction, which causes BAX translocation to the mitochondria. Additionally, RASSF1A was shown to regulate BAX activity through MST1 [66]. In this case, MST1 directly phosphorylates the antiapoptotic BH3 protein BCL-xL allowing for BAX translocation to the mitochondria. The direct regulation of the tumour suppressor p53-MDM2 complex is another clear link between RASSF1A signalling and the canonical apoptotic pathways [55]. Concomitantly, RASSF1A binds to the DR associated protein DAXX, which results in the disruption of the interaction between MDM2 and HAUSP, resulting in MDM2 auto-ubiquitination and degradation inducing p53 stabilisation, which leads to cell cycle arrest and apoptosis.

#### 3.2.2. Hippo Pro-Apoptotic Pathway

The kinases MST1 and MST2 were two of the first described RASSF1A interactors [67]. These kinases are the homologues of Drosophila Hippo kinase that give the name to the pathway. LATS1/2 kinases are the major effectors of MST1/2, which inactivate the transcriptional activity of YAP/TAZ proteins, according to the canonical view of the pathway [6]. Nevertheless, early experiments showed that RAS promotes the interaction of RASSF1A with MST1/2, resulting in the activation of MST1/2 kinase activity and triggering apoptosis [51,67]. Conversely, the inhibitory binding of RAF1 to MST2 prevents activation and apoptosis induction. These interactions were shown to be regulated by DR and RTKs [51,59]. We showed that upon pro-apoptotic conditions, RASSF1A scaffolds the MST2 interaction with LATS1 facilitating the phosphorylation of LATS1 by MST2, which leads to YAP1 phosphorylation and its nuclear translocation [51]. In this case, YAP1 interacts with p73, leading to the formation of the YAP1-p73 complex and the transcription of pro-apoptotic genes, when the pathway is activated by RASSF1A. Moreover, RASSF1A also mediates the activation of other effectors of the Hippo pathway through the core kinases MST1/2 and LATS1/2. For instance, oncogenic KRAS promotes the stabilisation of p53 in a RASSF1A dependent fashion in colorectal cancer [4]. In this case, active LATS1 binds to MDM2 preventing the ubiquitination and degradation of p53. The O’Neill group has shown that ATM/ATR activates the Hippo pathway in a RASSF1A dependent fashion regulating p73 dependent transcription [39]. This group also showed that, upon DNA damage, ATM/ATR phosphorylation of RASSF1A promotes LATS1 inhibition of CDK2, leading to the stabilisation of DNA replication forks [64]. Importantly, different groups have shown that RASSF1A can regulate Hippo member functions without engaging the full pathway. For example, RASSF1A regulates YAP1 activity downstream of TGFβ by restricting the interaction of YAP1 with SMAD2, and TGFβ induced RASSF1A degradation is necessary to allow for the formation of the YAP1/SMAD2 complex and its nuclear translocation [48]. Furthermore, RASSF1A can counteract the YAP1 mediated transcriptional repression of tumour suppressor genes in a p53 dependent fashion [68]. RASSF1A can also mediate the direct phosphorylation of H2B by MST2, which is required to allow for repair in response to DNA damage [69]. Finally, the RASSF1A-Hippo module seems to regulate, and be closely regulated by, the RAF-ERK and AKT pathways though different crosstalk mechanisms [4,60,70,71]. Importantly, several members of the Hippo pathway are commonly deregulated in tumours, which, together with the high rate of loss of expression of RASSF1A, indicates that this module of the RASSF1A SN plays a fundamental role in the regulation of cellular homeostasis.

#### 3.2.3. Cell Cycle Regulation

The role of RASSF1A in cell cycle regulation is well established and several proteins have been shown to mediate this effect [5]. In general, RASSF1A is an inhibitor of cell cycle progression, explaining why its loss of expression could lead to cell transformation. This role of RASSF1A is related to the regulation of CDK-cyclin complexes through a plethora of interactors [5]. Several PTMs regulate and coordinate RASSF1A-dependent signalling during the different cell cycle phases. One of the very first RASSF1A effectors described was the APC/CDC20 complex, which is a multiprotein complex that controls the transition from metaphase to anaphase by triggering the degradation of S-phase and M-phase cyclins and the cohesion complex, thereby allowing for sister chromatid separation to ensue [72,73]. RASSF1A inhibits APC/CDC20 activation by binding to CDC20, thus maintaining high CDK1 activity and prometaphase arrest [46,74]. The phosphorylation of RASSF1A by Aurora A/B kinases converts RASSF1A from an APC/CDC20 inhibitor to a substrate, which results in RASSF1A degradation and mitotic progression [46]. RASSF1A was also shown to regulate cell cycle progression through interacting and enhancing the activity of the transcription factor p120 E4F, a repressor of Cyclin A expression [75,76]. RASSF1A also suppresses cell cycle progression through inhibiting its direct interactor JNK, resulting in cyclin D1 downregulation, the accumulation of the p27^Kip1^ CDK2/4 inhibitor, and cell cycle arrest [77,78].

Another link of RASSF1A signalling with cell cycle regulation is its role as a modulator of microtubule dynamics [79,80,81]. Interaction with microtubules was the first described function of RASSF1A. Subsequently, several studies have shown that RASSF1A interacts with MAP1B and MAP1S/C19ORF5 and it dynamically associates with different microtubular structures, including the cytoskeleton and the mitotic spindle [79]. During mitosis, RASSF1A localisation changes from the centrosome to the spindle poles and the mid body [82]. RASSF1A seems to regulate the mechanical properties of these structures through direct interaction with the microtubules by a basic region between amino acids 120–288 [66]. The stabilisation of microtubules is an important property for preserving genomic stability and the tumour suppressor function of RASSF1A [80]. Interestingly, several of RASSF1A regulators and effectors, such as Aurora A, PKC, MST2, and LATS1, also localise to these structures, indicating that some modules of the RASSF1A SN could also be regulated during these mitotic processes to secure the correct segregation of chromosomes [5,82]. 

#### 3.2.4. WNT/β-Catenin Pathway

The WNT/β-catenin pathway plays an important role in the regulation of cellular proliferation and this pathway is often de-regulated in cancer [83,84]. β-catenin is a protein with a dual function as a cell adhesion molecule in complex with cadherins and as a transcription factor when released from cadherin. Several PTMs, including phosphorylation by GSK3-β and ubiquitination by the SCF^β-TrCP^ complex, followed by degradation, regulated free β-catenin. The activity of this β-catenin destruction complex is blocked by WNT ligands that allow accumulation of β-catenin and its subsequent translocation to the nucleus, where it activates the expression of proliferation genes [83,84]. Different reports have shown that β-catenin can be regulated by RASSF1A [6]. RASSF1A increases GSK3-β-mediated β-catenin phosphorylation, leading to enhanced β-catenin degradation and the inhibition of the WNT/β-catenin pathway [85]. Conversely, GSK3-β can phosphorylate RASSF1A, which prevents its binding to and activation of the MOAP-1 apoptosis pathway [41]. It is worth noting that the Hippo and WNT pathways crosstalk through their transcriptional effectors YAP and β-catenin associating to transcribe stemness genes. RASSF1A expression diverts YAP from this complex to bind to p73, leading to the transcription of differentiation genes [86]. 

#### 3.2.5. Rho Family Signalling Pathways

A role of RASSF1A in the regulation of Rho signalling has also been shown. Rho signalling is mediated by the small GTPases Rho, Rac, and CDC42 and it is involved in the regulation of cell migration, transcription, and cytokinesis [87]. This module is a key regulator of metastasis and RAS dependent transformation with RhoA promoting these processes while RhoB inhibits them [88,89]. Different groups showed that RAS negatively regulates Rho signalling through RASSF1A. For instance, RhoA is a direct interactor of RASSF1A and promotes Smurf1-dependent RhoA ubiquitination, leading to RhoA degradation, and, as a consequence, prevents oncogenic Rho functions [90]. Moreover, RASSF1A can modulate the activity of Rho family of guanidine exchange factors, such as VAV2, ECT2, and GEFH1, which promote the activation of Rho proteins [91]. In particular, several groups have shown that RASSF1A regulates the activity of GEFH1 [91,92,93]. In fact, RASSF1A induces the dephosphorylation of the inhibitory S885 in GEFH1, which in turn allows for GEFH1 to activate the anti-metastatic RhoB GTPase [91]. NDR2, the kinase phosphorylating and suppressing GEFH1, is inhibited by RASSF1A, which results in the stimulation of RhoB activity [93]. RhoB also enhances the NDR2 mediated phosphorylation of YAP, which prevents YAP from translocating to the nucleus and regulating transcription [91,93]. Through this signalling module, RASSF1A can regulate cell migration and invasion, rationalising why the loss of RASSF1A expression can lead to metastasis. 

#### 3.2.6. Other Modules of the RASSF1A Signalling Network

Ever growing evidence has shown that RASSF1A can regulate other well-known signalling pathways. Among them is the aforementioned role of RASSF1A in the regulation of the hypoxia response. Dabral et al. [43] performed a comprehensive mechanistic characterisation of how RASSF1A and HIF-1α regulate each other in the context of pulmonary hypertension. HIF-1α induces *RASSF1A* transcription through direct binding to the *RASSF1A* promoter. Under hypoxic conditions, PKC phosphorylates RASSF1A, leading to increased RASSF1A stability and promoting RASSF1A-HIF-1α interaction, which prevents HIF-1α degradation. Further studies are necessary for fully understanding the physiological role of this positive feedback loop, but the observation that *RASSF1A* knockout animals do not show pulmonary hypertension indicates that this feedback is a pathophysiological function of RASSF1A. 

In addition to mediating the DNA damage response, RASSF1A might be involved in the DNA repair machinery. RASSF1A forms a complex with the xeroderma pigmentosum A (XPA) DNA repair protein upon DNA damage and regulates the XPA acetylation-deacetylation cycle by recruiting the deacetylase SIRT1 to its substrate XPA upon UV treatment [94]. In this module, RASSF1A promotes DNA repair together with the stabilisation of the replication forks [64]. Importantly, the RASSF1A A133S SNP variant inhibits XPA deacetylation, stabilises a RASSF1A-XPA complex, and decreases DNA repair [94]. Interestingly, the RASSF1A A133S SNP variant has been identified in a range of cancers [95], and its association with mutations in *BRCA1/2*—two other DNA repair genes—might indicate the inhibition of DNA repair as pathogenetic basis for the cancer association of this variant. 

Finally, it must be stressed that, as already illustrated in the previous sections, different crosstalk mechanisms link RASSF1A with the RAF-ERK and AKT pathways. Our view is that it is impossible to understand RASSF1A signalling and its role in cancer isolated from the signalling network that includes these RAS effectors. Some of these crosstalks are related to the regulation of the Hippo pro-apoptotic signalling. RAF1 interaction regulates free β-catenin, but we have also shown that MST2 and LATS1 regulate the RAF1-MEK1 interaction and signalling in a RASSF1A dependent manner [60,70]. Importantly, while wild type BRAF does not bind to MST1, the oncogenic mutant BRAFV600E can bind to this tumour suppressor, which could conceivably shut down the anti-oncogenic signal mediated by MST1/2 in several cancer types [96]. Additionally, AKT inhibition of MST1/2 by direct phosphorylation is prevented by RASSF1A interaction with MST2 [60,97]. Conversely, MST1 was shown to phosphorylate and inhibit AKT [98], which has been related to a possible role of RASSF1A in the activation of autophagy and suppression of oncogenic transformation in HCC [99]. Interestingly, a role of AKT in the induction of *RASSF1A* promoter hypermethylation has also been proposed [100], which could constitute an interesting intercalated negative feedback loop, where AKT1 and MST1/2 can mutually inhibit each other, with AKT, in addition, being able to suppress the expression of the MST1/2 activator RASSF1A that is relevant for the pathological silencing of this tumour suppressor. The regulation of RASSF1A by GSK3β, as mentioned above, is another point of crosstalk among these modules [41]. Although not well characterised, CNK1, which is another scaffold protein, might be key in coordinating cross-regulation among these pathways [101]. CNK can bind RASSF1A, MST2, AKT, and RAF proteins, which constitutes a physical interaction platform for these different pathways, but the RASSF1A mediated regulation of the ERK and AKT modules by these complexes might be MST1/2 independent. 

## 4. RASSF1A SN Druggability: Possible Points of Interference and Challenges

The role of RASSF1A in cell transformation is widely accepted and the loss of expression of this protein is one of the most common events in solid tumours [1]. Therefore, trying to restore the normal function of RASSF1A in cancerous cells that have lost its expression could lead to new therapeutic options for a wide range of cancer types. In the following section, we summarise some of the therapies that have been tested to treat RASSF1A defective tumours and propose some strategies for targeting RASSF1A SN nodes. However, it must be highlighted that targeting a scaffold is not an easy task and several approaches must be considered when exploring the possible strategies to target the RASSF1A SN for cancer therapy. The main challenge is that RASSF1A has no known enzymatic activity and epigenetic modifications generally cause its loss of function. Based on these observations most efforts to restore RASSF1A tumour suppressor function are focussed on targeting the epigenetic machinery that mediates *RASSF1A* DNA promoter methylation. Demethylating therapies are already used in cancer therapy and several new drugs are at different stages of clinical development [102]. Some of these agents are well accepted as chemo-sensitizers and it is possible that part of this effect is due to re-activating RASSF1A expression in cancer cells. However, the success of these treatments in RASSF1A deficient tumours might be limited by concomitant mutations of genes that are part of the RASSF1A SN, which we have delineated here. For this reason, a putative point of intervention could be developing RASSF1A mimetic agents that can rescue RASSF1A functions. The challenge with RASSF1A mimetics would probably be that, as a scaffold, most of its functions are mediated by protein-protein interaction and it is not clear how one could substitute these functions with drugs. 

A more successful opportunity to rescue RASSF1A deregulation in cancer could come from the knowledge that we have of RASSF1A SN. The loss of expression of RASSF1A leads to the activation of pro-oncogenic signals mediated by several of the modules of the network. Importantly, we already have an arsenal of targeted drugs for some of the nodes of this SN that are deregulated in cancer, such as CDKs, Aurora kinases, RAFs, and AKT. Hence, the development of effective treatments against tumours that have lost the normal RASSF1A SN function might require multi-targeting of different key network nodes that are deregulated when RASSF1A function is lost (Figure 3). Based on our current knowledge of the RASSF1A SN, we have to consider that some of its tumour suppressor functions could be inhibited, even in tumours that retain RASSF1A expression. Such inhibition could be mediated by two mechanisms; (i) PTMs that affect the affinity of RASSF1A for its effectors (phosphorylation) or promote degradation of this scaffold (ubiquitination); and, (ii) the deregulation of the expression or activity of nodes in the RASSF1A effector SN. Taking this into account, it is possible that we can develop several new strategies that aim to rescue RASSF1A loss of function by developing targeted or combination therapies that consider the whole RASSF1A SN.

## 5. Therapeutic Intervention

### 5.1. Inhibition of DNA Methylation 

#### 5.1.1. Non-Nucleoside DNA Methyltransferase Inhibitors

The development of DNMT inhibitors (DNMTi) for the treatment of tumours is a hot topic of research in drug development (for a thorough review see [102]). To date, there are two such drugs that are approved for the treatment of haematological diseases and leukaemias, 5-azacytidine (AZA), and 2’-deoxy-5-azacytindine (5-aza-CdR). These drugs are nucleoside analogues that can be incorporated into DNA and—when methylated by a DNMT—produce a covalent DNMT-DNA complex that inactivates DNMTs by subsequent degradation of the trapped enzymes and also initiates DNA damage [103]. In vitro and in vivo data have shown that AZA and 5-aza-CdR mediate the re-expression of RASSF1A in different cell lines [104]. Most of these studies confirm that the re-expression of RASSF1A results in the reactivation of apoptosis and RASSF1A mediated cell cycle inhibition. For instance, AZA treatment of retinoblastoma HXO-RB44 cells promotes the expression of RASSF1A and the induction of RASSF1A dependent apoptosis [105]. Moreover, in non-small cell lung cancer cell lines, the re-expression of RASSF1A induced by AZA and 5-aza-CdR treatment arrested cell cycle progression in G0/G1 phase and inhibited cell migration [106]. In mouse models of uveal melanoma, where UM-15 cells express RASSF1A after treatment with AZA, the formation of intraocular tumours was prevented and the development of subcutaneous tumours was delayed [107]. A serious potential problem with DNMTi is that they cause widespread demethylation of genes, which can lead to the re-expression of tumour suppressor and cancer cell death genes, but can also activate the expression of oncogenic genes. This was illustrated by observations in MCF7 cells, where the re-expression of RASSF1A led to activation of apoptosis, but also to a simultaneous expression increase in a series of pro-metastatic genes [107]. This might prevent the use of these agents as single agents, but they are expected to be effective as part of combination therapy. This problem could be alleviated by using more selective compounds at low concentrations that help in sensitising the cell to other drugs. For instance, SGI-110, a second generation 5-aza-CdR pro-drug that is already being tested in clinical trials [108,109], was shown to promote RASSF1A expression and prime ovary and testis cancer cells to respond to cisplatin treatment [110,111].

#### 5.1.2. Other DNMT Inhibitors

Other types of DNMTi target the enzymatic activity of DNMTs by binding to the catalytic region and they are known as non-nucleoside analogues [107]. These types of DNMTi have not been tested yet in clinical trials and there is very little information relating them to the specific de-repression of RASSF1A gene expression. Several strategies have been proposed to target these enzymes, in addition to direct inhibitors of DMNTs. One of them is targeting metabolic pathways that are necessary for the modification of DNA by DMNTs. In this strategy, it has been proposed that the Methionine Cycle could be inhibited, which produces S−adenosyl methionine, a major donor of methyl groups in the cell. This type of inhibitors includes classical anti-folates such as methotrexate, which is one of the oldest chemotherapeutic agents, but also new antifolates. Promisingly, TMECG, which is one of the novel-antifolate agents, mediates RASSF1A promoter demethylation in melanoma and breast cancer cells, especially when used in combination with dipyramidole (DIPY) [112]. These compounds prevent the synthesis of S-adenosyl-methionine, the substrate of DNMTs, which in turn results in a decrease in DNA methylation.

#### 5.1.3. Natural Agents

The natural anti-oncogenic compounds are one area of drug discovery where RASSF1A re-expression has been related to possible anti-cancer therapy and even cancer prevention. Although many of these studies need follow up research in appropriate pre-clinical models, a striking number of these natural compounds seem to act, at least in part, by inhibiting DNMTs and promoting RASSF1A expression (for a review see [104]). None of these compounds are thought to act as cytidine analogues, but they are thought to mediate their effects on DMNT through a different mechanism of actions. One such mechanism is the regulation of metabolic routes, as already illustrated by the anti-folate inhibitors that are mentioned above. Thus, there is evidence that folate could lead to the increase in RASSF1A promoter methylation [113]. However, other natural methyl donors, such as methionine and vitamin B12, were shown to reduce RASSF1A promoter methylation, which indicates that these compounds might have different effects on DNA methylation [114]. Another group of natural compounds seems to work as DNMT activity inhibitors that target different regulators of these proteins through several mechanisms of action. First, mahanine, which is a natural antioxidant agent that is present in the curry tree, was shown to induce RASSF1A expression [115,116]. Mechanistically, mahanine seems to induce proteosomal degradation of DNMTs by preventing the stabilising phosphorylation of these enzymes by AKT. Mahanine was shown to inhibit PDK1, which is a kinase that phosphorylates and activates AKT downstream of PI3K [116]. On the other hand, direct inhibition of DNMT has been observed to be mediated by peperomin E, a polyphenol, which results in the expression of RASSF1A, RUNX3, APC, and p16INK4 in lung cancer cells and mouse lung tumours [117]. This compound can bind the active site of DNMT1, which inhibits its activation and promotes its degradation. Importantly, DNMT degradation by natural compounds has been shown as mechanism of action of several natural compounds that promote RASSF1A expression in cancer cells. For instance, the phytochemical PEITC reduces the expression of DNMT1, DNMT3A and DNMT3B, resulting in the induction of RASSF1A dependent apoptosis in lung cancer cells [118]. Additionally, a group of polyphenols has been shown to mediate DNMT degradation, although the actual mechanisms by which they achieve this effect have not been properly studied. This group includes curcumin, emodin, and resveratrol, and they can also promote the expression of other tumour suppressor genes that are downregulated by DNA methylation [119,120,121]. Interestingly, it has also been reported that, in rats fed with olive oil, DMNT activity decreases with a concomitant reduction of *RASSF1A* and *TIMP3* gene methylation, while corn oil promotes DMNT activity and DNA methylation [122]. Finally, 3,3-dindolylmethane (DIM) was shown to increase the expression of RASSF1A and promotion of MST1/2-LAST1 complex in gastric cancer cells, although it has not been shown that this is caused by the RASSF1A promotor DNA demethylation [123]. Altogether, these reports show the ability of natural compounds to prevent and reverse RASSF1A gene methylation. The characterisation of the mechanism of action of these compounds might lead to new avenues for the treatment of RASSF1A defective tumours. 

### 5.2. Inactivation of the Hippo Pathway and How This Can Be Reverted

Of all the modules of the RASSF1A signalling network, the Hippo/MST pathway has been most clearly associated with the tumour suppressor activity of this protein. Thus, the RASSF1A loss of expression leads to a shutdown of pro-apoptotic signalling by the Hippo pathway while promoting YAP1 proliferative signalling [6]. Additionally, the loss of RASSF1A pro-apoptotic signalling can occur by the loss of expression of MST1/2 and LATS1/2 in some cancer types. Hence, a possible strategy to rescue the loss of the RASSF1A pro-apoptotic signal would be to reactivate the core kinases of the Hippo pathway or prevent YAP1 pro-oncogenic signalling. In this vein, it has been proposed that the disruption of the YAP1-TEAD interaction would prevent the transcription of oncogenic genes [123]. Different studies have shown that verteporfin, a drug that is approved for the treatment of macular degeneration, prevents YAP1-TEAD dependent transcription. In several cell lines, this photosensitiser shows an anti-oncogenic effect that is related to a decrease in YAP1-TEAD activity [124,125,126]. Verteporfin was shown to sensitise triple negative breast cancer cells to taxol and radiation, allowing for the activation of DNA damage-induced apoptosis [127]. This drug is now widely used as an inhibitor of YAP1 oncogenic signalling in preclinical experiments [126], and clinical trials have commenced for prostate, breast, and pancreatic cancer [128]. Of note, no study has tested the relationship between verteporfin treatment and RASSF1A expression, but it would be very informative to see whether verteporfin could rescue the loss of RASSF1A. Importantly, additional drug screens identified statins, dasatinib, and pazopanib, as potential inhibitors of YAP1 oncogenic signalling [129,130]. We should expect to see information soon whether YAP inhibition is part of the mechanism of action of this compounds since these drugs are already FDA approved. It would be very informative to test whether there is any correlation between response to these treatments and RASSF1A expression, based on the knowledge that we have of the RASSF1A-Hippo pathway. 

Theoretically, the activation of the MST1/2 and LATS1/2 pro-apoptotic signal in the absence of RASSF1A could lead to the rescue of the anti-oncogenic signal that is mediated by RASSF1A. However, it would be very challenging to develop drugs that activate these kinases and they could potentially lead to severe side effects. Nevertheless, there might be ways to induce the activation of these kinases by targeting other proteins in the RASSF1A SN. For instance, a recent report has shown that, in enterocytes, MST2 signalling is negatively regulated by FGFR4 in response to an increase in the concentration of intestinal bile acids [131]. This work showed that the inactivation of MST2 in response to bile acid is caused by inhibitory interaction with RAF1. Moreover, the authors showed that, in animal models, the depletion of bile acid metabolism by drug treatment could lead to the activation of MST1/2 and prevent oncogenesis. Although this work did not test if RASSF1A could be affected by bile acids, the observation serves as a proof of principle that MST1/2 can be reactivated in some tumours. Critically, this work indicated that disrupting the RAF1 inhibitory effect on MST2 could be a potential treatment for some cancers, providing the opportunity to develop RASSF1A mimetics. One possibility would be to develop small molecules that can interfere the inhibitory interaction of MST1/2 with RAF proteins. In fact, work from our group showed that a small amino-terminal stearoylated MST2 peptide disruptor could be used to disrupt the MST2-RAF1 interaction and affect the normal development of the heart of Zebrafish [70]. It is tempting to speculate that RASSF1A mimetics could be a treatment for tumours harbouring this oncogene while taking into account that BRAFV600E binds and inhibits MST1/2 in cancer cell lines.

### 5.3. BAX Mimetics

Another effector module that can be targeted to rescue the loss of expression of RASSF1A is the DR-MOAP-1-BAX pathway. As explained above, RASSF1A can induce the activation of the pro-apoptotic BAX protein via MOAP-1 [65]. The loss of RASSF1A expression also contributes to mechanisms of resistance to TRAIL treatment [132]. Additionally, the prevention of BAX oligomerization and translocation to the mitochondria is commonly observed in cancer cells. BAX is part of the BH3 family of pro-apoptotic proteins and it is negatively regulated BCL2 and other anti-apoptotic member of this family. For this reason, different BH3 mimetics have been developed that prevent the interaction of BCL2 with BAX [133]. Such drugs are highly effective in the treatment of multiple myeloma [134], although it is unknown whether the response is related to RASSF1A expression status. Interestingly, a small study of Chinese multiple myeloma patients showed frequent RASSF1A hypermethylation [135]. Thus, BH3 mimetics should be evaluated as an option to treat RASSF1A deficient tumours. 

### 5.4. Perspective: Multitargeting of the RASSF1A Signalling Network

The last two decades have seen the development of a plethora of drugs that are designed to specifically target single proteins that are the main drivers of cancer development. Although some of these agents are now widely used in the clinic and they are invaluable additions to the arsenal of treatments for cancer, many of these agents have failed to live up to expectations. In parallel, we have increased our knowledge regarding SNs and the importance that rewiring of these networks plays in the lack of response and development of resistance to targeted therapies [136]. Based on this evidence, it is worth considering that we need to combinatorially target several nodes in its SN in order to successfully rescue the tumour suppressor activity of RASSF1A. One thing that we must consider is that RASSF1A signalling function might also be lost in those tumours that still express this protein. It is common that cancer cells keep intact pro-apoptotic machinery, but the cells lose sensitivity to pro-apoptotic signals [137]. In the case of tumours that express RASSF1A, this sensitivity could be lost by the hyper-activation of the proteins that regulate its PTMs. Moreover, aberrant PTM of RASSF1A might result in incomplete rescue of its tumour suppressor function, even if we could restore RASSF1A using DNMTi. In both scenarios, targeting one or several of the enzymes that negatively regulate RASSF1A anticancer properties might lead to successful treatments. Taking this into consideration, we may already have several drugs that are available to be used in combination with other agents to reactivate RASSF1A function. For instance, CDK4/6 inhibitors are already approved for the treatment of breast cancer [138,139]. These drugs prevent hyper-phosphorylation of the Retinoblastoma 1 protein and they might prevent the CDK4 inhibitory phosphorylation of RASSF1A favouring proliferation. Similarly, targeting Aurora A and PKC would prevent the negative effect of these proteins on RASSF1A. Aurora kinase inhibitors are being currently tested in clinical trials with limited efficacy for solid tumours, but they are promising agents for the treatment of hematologic malignancies, especially in combination with other kinase inhibitors [140]. It would be interesting to test whether RASSF1A expression status might contribute to the low efficacy of these drugs and whether re-expression of RASSF1A could increase their efficacy. Another RASSF1A regulating kinase that is the object of intensive research as a therapeutic target for several diseases is GSK3-β. However, this protein might function as a tumour suppressor or oncogene in different cancer types, and the use of GSK3-β inhibitors for cancer treatment needs further study [140]. Finally, TGFβ inhibitors that have also been tested in clinical trials might prevent RASSF1A degradation in certain cancer types [141]. Another strategy for restoring RASSF1A physiological expression could be preventing its ubiquitination by ubiquitin ligases. The rationale for developing E1, E2, and E3 ligases inhibitors is strong and several drugs have been tested in preclinical models, but they are yet to be tested in the clinic [142]. However, proteasome inhibitors, such as bortezomib, are already used for cancer treatment and they may help rescuing the expression of RASSF1A. Intriguingly, another member of the RASSF family, RASSF4 re-expression has been shown to potentiate the effect of bortezomib in multiple myeloma cell lines when this drug is used in combination with MEK1/2 inhibitor [143], which indicates that the same might occur for RASSF1A.

Importantly, while looking at the topology of the RASSF1A SN, it is clear that the AKT and RAF-ERK pathway closely regulates the tumour suppressor activity of this protein. These pathways are activated in most cancer cells and combination therapies for different cancer types have shown positive results [144,145]. Several lines of evidence show that RASSF1A expression may increase the effectiveness of these treatments and even prevent the acquisition of resistance to these treatments. On the other hand, a combination of DNMTi that reinstate RASSF1A expression in combination with RAF inhibitors may lead to the activation of apoptosis in BRAF mutant tumours. Another possibility that could be explored is whether inhibition of EGFR oncogenic signal might lead to the activation of the RASSF1A pro-apoptotic signal downstream of mutant KRAS [59].

Finally, the pharmacological inhibition of RASSF1A promoter hypermethylation could result in an increase of the response to death receptor treatments. For instance, TRAIL agonists have been tested in clinical trials, but they have shown low efficacy as single agents, and many patients developed resistance [146]. This resistance might be caused by the downregulation of RASSF1A expression and prevention of the activation of MOAP-1-BAX and Hippo pathway pro-apoptotic signals [53]. For this reason, the combination of TRAIL agonist treatment with treatments that lead to the expression of RASSF1A could increase the effectiveness of TNF targeting therapy.

## 6. Conclusions

There is much that we still have to learn about the physiological role of RASSF1A in signal transduction and how its deregulation contributes to cell transformation. However, restoring the normal regulation of its signalling network, or at least the modules that lead to cell transformation, could lead to the development of novel therapeutic options with improved outcomes for several cancer types. 

## Figures and Tables

**Figure 1 cancers-12-00229-f001:**
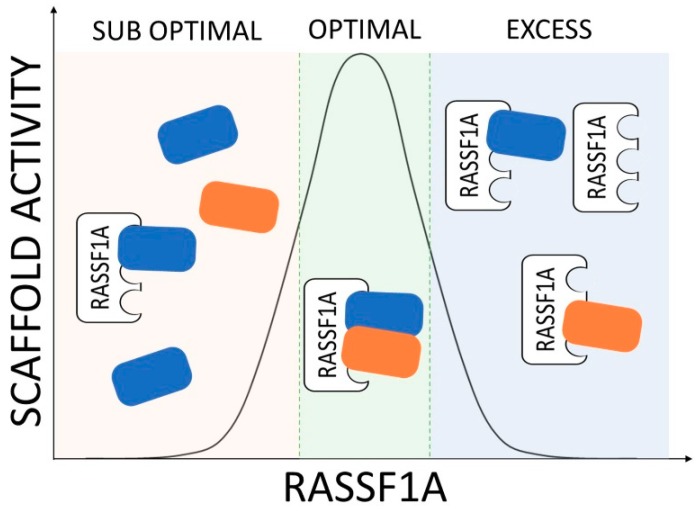
RASSF1A signalling is dependent on its level of expression and the stoichiometry of its complexes. As exemplified by the RASSF1A-scaffolded complex (blue and orange boxes), if there is no RASSF1A expression, or sub optimal expression, there is low complex formation and the signal will be low (left); in the case of intermediate RASSF1A expression, signal will be high (centre), this is the physiological range of expression; if there is too much RASSF1A the signal will be low (right).

**Figure 2 cancers-12-00229-f002:**
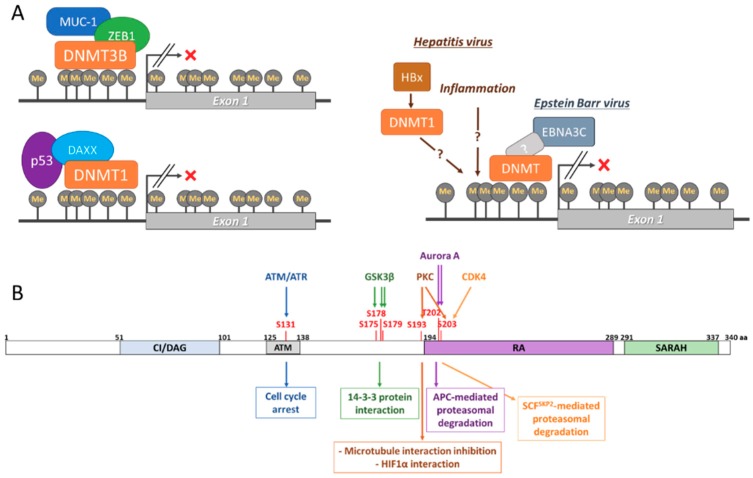
(**A**). Epigenetic regulation of *RASSF1A* expression. DNA hypermethylation of the *RASSF1A* promoter mediated by DNMT1 and DNMT3 complexes associated with different proteins including MUC-1-ZEB1 and DAXX-p53 (left) and viral proteins (right). (**B**). Schematic of RASSF1A protein structure. RASSF1A contains different domains: C1/DAG (diacylglycerol), ATM (ataxia-telangiectasia mutated) domain, RA (Ras association) domain and SARAH (Salvador-Hippo-RASSF) domain. Several residues are phosphorylated (red) by the indicated kinases regulating the degradation of the RASSF1A protein and the mediation of different biological functions by this tumour suppressor.

**Figure 3 cancers-12-00229-f003:**
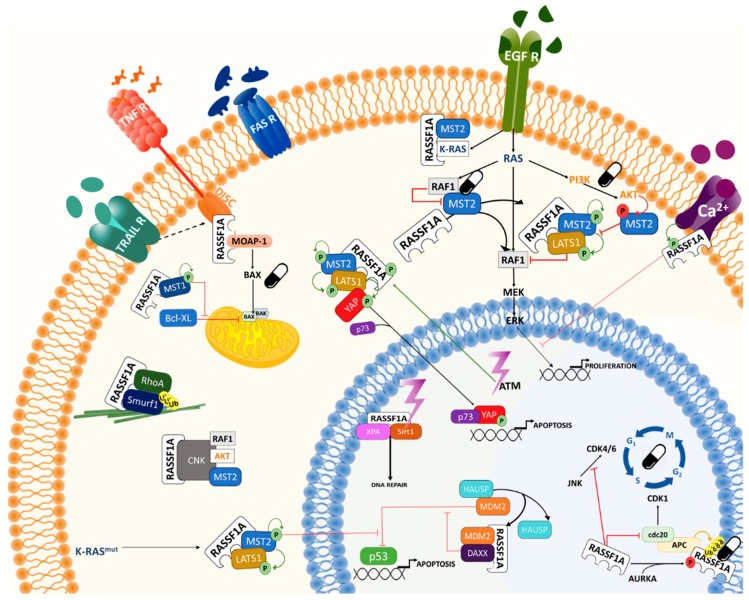
The RASSF1A signalling network is regulated by the death receptors and receptor tyrosine kinases. The signalling network includes the cell cycle, Hippo, p53, MAPK, AKT, Rho, and apoptotic signalling modules. The pill icons indicate possible nodes to target for therapeutic intervention in cancer.

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
