# Peer review of "RASSF1A Tumour Suppressor: Target the Network for Effective Cancer Therapy"

_cancers, 2020, doi:10.3390/cancers12010229_

Round 1

Reviewer 1 Report

The review written by García-Gutiérrez and colleagues, 2019, under consideration for publication has presented a thorough and thoughtful overview of the RASSF1A tumour suppressor. The article includes a short introduction regarding RASSF1A function, different mechanism of regulation of RASSF1A and its complex signalling network. The review progresses clearly into key therapeutic areas including potential druggability.

Please find immediately below very minor items that the authors may address prior to final publication.

Manuscript Strengths:

This is a thoughtful and very comprehensive review. Figure 2 and 3 illustrating the RASSF1A epigenetic regulation, protein structure and SN are excellent.

Manuscript Areas of Focus and Additional Clarification: 

In some areas of the text it is not clear whether a certain or multiple tumour types are mentioned. For example, on page 4, 128-129 (phosphorylation of S131): It is mentioned “…seems to increase resistance to radiotherapy in some patients”. However, the article reference refers to soft tissue sarcoma. It would be helpful to clarify in the text whether it is seen in sarcoma or across multiple tumour types. Page 13, 538-539 (CDK4/6 inibitors): “…approved for the treatment of solid tumours”. Solid tumours is very broad, consider rephrasing it to current approval (breast cancer). Reference 128 appears incomplete: “Availabe online: https://www.cancer.gov/about-cancer/treatment/clinical-trials/intervention/verteporfin (accessed on”. Authors may also consider mentioning tumour types where its being tested (prostate, breast, pancreas). Given the comprehensiveness of the review, the authors may include more content in the concluding paragraph.

Author Response

We want to thank both reviewers for their positive comments. We have now incorporated most of the changes suggested and made additional editions to the text to improve the style and presentation. Please see bellow our reply (in blue) to the reviewer suggestions.

The review written by García-Gutiérrez and colleagues, 2019, under consideration for publication has presented a thorough and thoughtful overview of the RASSF1A tumour suppressor. The article includes a short introduction regarding RASSF1A function, different mechanism of regulation of RASSF1A and its complex signalling network. The review progresses clearly into key therapeutic areas including potential druggability.

Please find immediately below very minor items that the authors may address prior to final publication.

Manuscript Strengths:

This is a thoughtful and very comprehensive review. Figure 2 and 3 illustrating the RASSF1A epigenetic regulation, protein structure and SN are excellent.

Manuscript Areas of Focus and Additional Clarification: 

In some areas of the text it is not clear whether a certain or multiple tumour types are mentioned. For example, on page 4, 128-129 (phosphorylation of S131): It is mentioned “…seems to increase resistance to radiotherapy in some patients”. However, the article reference refers to soft tissue sarcoma. Changed to indicate that the evidence is for this tumour type (line 137). We have made this correction and minor changes in other areas to reduce redundancy.

It would be helpful to clarify in the text whether it is seen in sarcoma or across multiple tumour types. Page 13, 538-539 (CDK4/6 inhibitors): “…approved for the treatment of solid tumours”. Solid tumours is very broad, consider rephrasing it to current approval (breast cancer). We have done this change to indicate that is for breast cancer (line 600).

 Reference 128 appears incomplete: “Availabe online: https://www.cancer.gov/about-cancer/treatment/clinical-trials/intervention/verteporfin (accessed on”. We have edited the reference.

Authors may also consider mentioning tumour types where its being tested (prostate, breast, pancreas). We have changed this a per suggestion (line 544)

Given the comprehensiveness of the review, the authors may include more content in the concluding paragraph. The review is rather long, and we have decided to make only the changed suggested by reviewer 2 to this section. We want to note that we have tried to include outlook throughout the text and a longer conclusion might be just an unnecessary repetition of information.

Reviewer 2 Report

This is a fine review by an established group of investigators. The review is comprehensive and will contribute to understanding the importance of RASSF1A in multiple convergent signaling pathways. I have one moderately important comment and several, very minor, grammatical catches that are easily addressed as follows:

Moderately important issue to address:

Figure 1 is unnecessary and does not convert very useful information. I recommend updating with known binding partners of RASSF1A in a particular or general system if possible, and if not, removing it. It would be much more informative to have a summary figure at the end of section 3 to bring together all of the mentioned signaling pathways with the RASSF1A hub, as was the central theme of the text and there was a lot going on in Figure 3, so a figure about those pathways in particular would be beneficial for understanding.

The list of very minor grammatical catches to easily fix:

“Activated MOAP-1 has been shown to activate “ too many activates in a row, it becomes confusing. line 197: “[63] might prove to be… “ is missing a noun before “might”. line 204: “A signalling network is regulated be the death receptors “ ---“be” should be “by” line 303: “Smurf1-dependnet RhoA”. Dependent is misspelled line 396: Most literature refers to deoxycitidine as 5-aza-CdR line 398-399: “in vitro” and “in vivo” should be italicized line 414: “SGI-110 a second “ there should be a verb or comma, something is missing after SGI-110 line 459 - Indoles (DIM) also affect RASSF1 https://www.spandidos-publications.com/10.3892/or.2013.2717 Closing statement: line 579: change: “could lead to the development of novel, improved therapeutic options for several cancer types” to “could lead to the development of novel therapeutic options with improved outcomes for several cancer types”

Author Response

We want to thank both reviewers for their positive comments. We have now incorporated most of the changes suggested and make additional editions to the text to improve the style and presentation. Please see bellow our reply (in blue) to the reviewer suggestions.

Reviewer 2

this is a fine review by an established group of investigators. The review is comprehensive and will contribute to understanding the importance of RASSF1A in multiple convergent signaling pathways. I have one moderately important comment and several, very minor, grammatical catches that are easily addressed as follows:

Moderately important issue to address:

Figure 1 is unnecessary and does not convert very useful information. I recommend updating with known binding partners of RASSF1A in a particular or general system if possible, and if not, removing it. It would be much more informative to have a summary figure at the end of section 3 to bring together all of the mentioned signaling pathways with the RASSF1A hub, as was the central theme of the text and there was a lot going on in Figure 3, so a figure about those pathways in particular would be beneficial for understanding.

We respectfully disagree that this figure does not add any information and were very keen in including this figure for researchers that are not very familiar with how scaffold proteins work. In fact, we think that this figure illustrates how we and others have used this knowledge about scaffold properties to study RASSF1A signalling. We have now changed the figure to include 2 different interactors that would be scaffolded by RASSF1A and extended the figure legend. We considered the suggestion of the reviewer of including known interactors, however we decided not to include specific names since this figure is in the introduction and specific proteins are not described yet. We also must point out that, as in the case of MST2 and LATS1 complex we do not have a complete idea of which proteins are direct interactors of RASSF1, i.e. only MST2 has been demonstrated to be a direct interactor, while LATS1 (despite strong evidence) has never been fully demonstrated to be a direct interactor and therefore we were worried that we could mislead the community.

The list of very minor grammatical catches to easily fix:

“Activated MOAP-1 has been shown to activate “ too many activates in a row, it becomes confusing. We changed to trigger “Activated MOAP-1 has been shown to trigger the extrinsic apoptotic module by regulating Bcl-2 proteins” (line 187)

line 197: “[63] might prove to be… “ is missing a noun before “might”. The noun is calcium signalling and we have added the commas to make that clear ”where calcium signalling, which is central in the regulation of myocyte physiology, [63] might prove to be a key regulator of RASSF1A function.”

 line 204: “A signalling network is regulated be the death receptors. Corrected

 “ ---“be” should be “by” line 303: “Smurf1-dependnet RhoA”. Dependent is misspelled line Corrected.

Most literature refers to deoxycitidine as 5-aza-CdR. We have changed this name throughout the text.

line 398-399: “in vitro” and “in vivo” should be italicized. Changed.

 “SGI-110 a second “ there should be a verb or comma, something is missing after SGI-110. Comma included (line 463).

 line 459 - Indoles (DIM) also affect RASSF1 https://www.spandidos-publications.com/10.3892/or.2013.2717 We have included this information and the reference in the section. It now reads “Finally, 3,3-dindolylmethane (DIM) was shown to increase the expression of RASSF1A and activation promotion of MST1/2-LAST1 complex in gastric cancer cells, although it has not been shown that this is caused by RASSF1A promotor DNA demethylation”.

Closing statement: line 579: change: “could lead to the development of novel, improved therapeutic options for several cancer types” to “could lead to the development of novel therapeutic options with improved outcomes for several cancer types”. We have done this change (line 642).
